# Material and Environmental Aspects of Concrete Flooring in Cold Climate

Jonny Nilimaa * and Vasiola Zhaka

Department of Civil, Environmental and Natural Resources Engineering, Luleå University of Technology, SE-97187 Luleå, Sweden
* Correspondence: jonny.nilimaa@ltu.se

**Abstract:** Dehydration of concrete floor slabs is a critical step to ensure that the flooring material adheres properly and that there is no moisture-related damage to the floor after installation. Dehydration in a cold climate is often a slow process, which can have a big impact on the overall duration of the construction project, and corresponding measures are often taken to accelerate the drying process, especially in constructions exposed to a cold climate. One common method, typically used to accelerate dehydration in cold weather, is to introduce internal heating cables into the slab. This method reduces the dehydration time, but may not be the best solution from a sustainability perspective. This paper presents a concept study of concrete flooring in a cold climate from a cradle to practical completion perspective. The study focused on the environmental and material aspects of the dehydration of concrete floors in a cast-in-place house. This paper showed that concretes with high water-cement ratios, which are typically preferred due to their low $CO_2$ emissions, may require measures for accelerated dehydration, which ultimately results in a higher environmental impact. The importance of environmental studies is also highlighted to fully understand the environmental aspects of construction.

**Keywords:** concrete; concrete construction; construction materials; dehydration; environmental impacts; LCA; EPD; flooring; cold climate construction

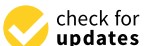



## 1. Introduction

*1.1. Moisture-Related Aspects of Concrete Flooring*

Drying, or dehydration, of concrete floor slabs is a critical step to ensure that the flooring material adheres properly and that there is no moisture-related damage to the flooring after installation [1]. Dehydration refers to the reduction of free water in the concrete and includes both chemical processes and evaporation. Dehydration in a cold climate is often a slow process, which can have a big impact on the overall duration of the construction project [2], and corresponding measures are often taken to accelerate the drying process, especially in cold climate construction [3]. If the dehydration process is not allowed to progress correctly, moisture-related problems can manifest in various ways and lead to expensive and time-consuming repairs [4]. If the concrete slab has a high moisture content, the flooring material can absorb the excess moisture, causing it to expand and buckle or warp [5]. Moisture in the concrete slab can also prevent the flooring adhesive from bonding properly, leading to adhesion failure and delamination of the flooring material. Moisture can bring minerals and other contaminants to the surface of the concrete slab, causing unsightly staining on the flooring material [6]. Moisture can create an ideal environment for mold and mildew growth, which can cause health issues and damage to the flooring material. Excess moisture can also lead to musty or unpleasant odors in the flooring material, which can be difficult to eliminate. To avoid these moisture-related problems, it is essential to properly manage the moisture content of the concrete slab before installing the flooring material. This can be achieved through

proper concrete curing, the use of moisture barriers or moisture-retardant systems, and appropriate environmental controls [7].

The requirements for dehydration of the concrete slabs depend on several factors, including the type and thickness of the slab, the type of concrete, and the environmental conditions during and after installation. The concrete slab should be dried until it reaches a moisture content of less than 5% to 6% or about 90% within the pores [8]. Moisture meters or moisture vapor emission rate (MVER) tests can be used to measure the moisture content of the slab. The drying time can vary depending on the thickness of the slab, the type of concrete used, and the environmental conditions. Generally, a minimum drying time of about 28 days after casting is usually recommended for a 100 mm-thick slab, to avoid moisture-related problems in the floor [9]. The temperature and humidity conditions during the drying process should be controlled to ensure optimal drying. The temperature is often recommended to be between 10 °C and 32 °C, and the relative humidity should preferably be between 40% and 60%. Adequate ventilation is essential to promote air circulation and moisture evaporation. Fans, dehumidifiers, and open windows can be used to improve ventilation. Before installing the flooring material, the concrete slab should be clean, smooth, and free of any contaminants such as oil, grease, or dust. It is important to note that these requirements may vary depending on the specific flooring material being installed. It is recommended that the construction workers consult the manufacturer's instructions and industry standards for specific drying requirements.

Special attention is always required for concrete construction in the fall or winter season in the Arctic region, and concrete drying is significantly affected by cold climate conditions [10]. Cold weather conditions can slow down the drying process and, in the worst case, even halt it altogether. The main reason for this is that cold temperatures slow down the chemical reaction that takes place during the curing process. These reactions are needed for concrete to harden and dry, and a cold climate can ultimately result in reduced concrete strength and increased moisture content. In extremely cold temperatures, concrete can freeze, which can lead to structural damage and delayed drying times. If the concrete freezes during the early stages of the curing process, it can weaken the final structure and lead to cracking and spalling. Generally, from a structural performance point of view, freezing must be avoided until the concrete has reached a compressive strength of at least 5 MPa [11]. Cold temperatures can cause moisture in the concrete to freeze, which can lead to structural damage and increased water penetration in the damaged structure. This can result in extended drying times and a higher risk of moisture-related problems, such as warping or cracking. Low temperatures can also cause the relative humidity levels of the air to increase, which can prolong the time it takes for the concrete to dry. High humidity levels can slow down the rate of moisture evaporation from the concrete, leading to extended drying times and potential moisture-related problems.

To mitigate the effects of cold climate conditions on concrete drying, it is essential to take appropriate measures during the curing process [12]. These can include using warm concrete, insulating blankets, or internal heating cables to keep the concrete warm. Other preventive measures include controlling the environmental conditions in the surroundings of the drying concrete slab, including external air heaters, regulating the humidity levels, and providing adequate ventilation to promote air circulation and moisture evaporation. Additionally, it may be necessary to extend the drying time of the concrete to account for the slower rate of drying in cold weather.

*1.2. Environmental Aspects of Concrete Flooring*

There are several environmental aspects that should be considered regarding concrete construction in general and concrete flooring especially. By considering these environmental aspects, it is possible to reduce the environmental impact of concrete flooring and make more sustainable choices.

Raw material extraction is one of the aspects that need to be considered in concrete construction [13–20] since it contributes significantly to the total environmental impact.

Concrete is made primarily of cement, sand, aggregates (which are extracted from quarries and mines), and additives. The environmental impact of these extraction processes should be considered, such as habitat destruction, energy use, and water consumption. Concrete is often produced off-site and transported to the construction site, which can result in greenhouse gas emissions from transportation. Choosing a local supplier or on-site mixing can, for example, reduce the transportation impact.

The material production constitutes a large part of the total environmental impact of concrete construction. Cement production is a major source of carbon dioxide emissions, which contribute to climate change. Using alternative materials, such as fly ash or slag, to replace some of the cement content can reduce the environmental impact considerably. Installation is an aspect that is sometimes forgotten when comparing different production alternatives for concrete flooring, especially for construction projects in a cold climate.

Concrete flooring may require energy-intensive processes for installation, such as external or internal heating alternatives. Concrete with a low cement content can result in lower material emissions, but it can also increase the need for energy-consuming heating alternatives to reach the dehydration limits within certain project timelines, as well as lower concrete strength. Choosing a more efficient installation method or using recycled materials for the sub-base can help to reduce the energy impact.

The need for maintenance should also be considered from a life cycle perspective. Regular maintenance of concrete floors may require the use of different chemicals, which can impact indoor air quality and contribute to environmental pollution. Choosing environmentally friendly cleaning products or implementing a maintenance plan that minimizes the use of chemicals can help to reduce the total impact of a concrete floor.

### 1.3. Analyzing the Environmental Impact of Concrete Floors

One way of analyzing the environmental impact of a certain product, for example a complete concrete floor, is to carry out an environmental product declaration (EPD) [21]. It is a document that provides transparent and verified information about the environmental impact of a product or service throughout its entire life cycle (i.e., a type of life cycle assessment (LCA)), from raw material extraction to the end of its useful life [22]. EPDs are typically based on international standards such as ISO 14025 [23] or EN 15804 [24], which provide guidelines for the development of Type III environmental declarations. EPDs provide quantitative data on a product's environmental impacts, such as energy consumption, greenhouse gas emissions, water consumption, and waste generation. EPDs can be useful for a variety of stakeholders, including architects, engineers, contractors, and building owners, as well as consumers and government agencies. They can help these stakeholders make more informed decisions about the environmental impact of the products they specify or use, and they can also support efforts to reduce the environmental impact of buildings and infrastructure.

The following description of EPDs is given by The Association for Swedish Concrete, Svensk Betong [25], which is a trade association for companies that manufacture ready-mixed concrete, practice concrete pumping, and manufacture and/or assemble concrete products:

> "Environmental Product Declaration is an international term for the environmental impact analysis declaration of a product. An EPD is a type III declaration, which means that it quantitatively (with numbers and data) describes the product's environmental impact during the entire life cycle (LCA based) and is produced according to one of the standards in the ISO 14040 series, alternatively based on the European standard EN 15804. It should be reviewed and verified by an independent third party and can be registered and published in e.g., the international EPD system, see also www.environdec.com. The declaration produced using Svensk Betong's EPD tool is therefore formally not an EPD but an environmental declaration with the same content and produced in the same way

as an EPD. Only when it is reviewed by a third party and registered does it become an EPD."

A recent investigation carried out by The Swedish Housing Agency, Boverket, published a report, "Development of rules on climate declaration of buildings" [26], where they recommended that limit values for climate impact should be introduced into the Swedish housing industry by 2027. Their analysis concluded that, to increase steering towards climate-improving measures being taken in planning and construction, limit values for climate emissions from buildings should be introduced in the Swedish construction industry by 2027. Further, they recommended that the limit value should cover the construction stage in the life cycle perspective, i.e., Modules A1–A5 (raw material supply in the product stage, transport in the product stage, manufacturing in the product stage, transport in the construction production stage, the construction and installation process in the building production stage). A study by Liljenström et al. [27] showed that almost 60% of an apartment building's total environmental impact comes from Modules A1–A5 and most of the remaining impact derives from Module B6 (operational energy use). Similar numbers for the environmental impact were, for example, declared for residential buildings in the UK by the Royal Institution of Chartered Surveyors (RICS) report on whole-life carbon assessment for the built environment [28].

The environmental factors analyzed in an EPD are typically [23,24]: global warming potential (GWP 100 years), ozone depletion potential (ODP), acidification potential (AP), eutrophication potential (EP), photochemical oxidant creation potential (POCP), and total primary energy. The following sections give brief descriptions of the various environmental factors.

### 1.3.1. Global Warming Potential (GWP 100 Years)

Global warming potential (GWP) is a measure of the ability of a greenhouse gas to contribute to the greenhouse effect and global warming. The scale is relative and compares the climate impact of greenhouse gases with the effect of the same amount of carbon dioxide. The emissions of different greenhouse gases can be converted to carbon dioxide equivalents using a GWP value, which makes it easier to compare them with each other. The GWP value is calculated according to the IPCC's method, where different greenhouse gases are referred to as the warming potential of carbon dioxide over 100 years [29]. Below, in Table 1, are two examples of greenhouse gases and their $CO_2$ equivalents.

**Table 1.** Common greenhouse gases and their corresponding $CO_2$ equivalents.

| Greenhouse Gas | $CO_2$ Equivalents |
|---|---|
| $CO_2$ | 1 |
| $CH_4$ | 28 |
| $N_2O$ | 265 |

### 1.3.2. Ozone Depletion Potential

Ozone depletion potential (ODP) is a measure of the potential thinning effect of chlorofluorocarbons (CFC gases) and other chemical compounds on the Earth's ozone layer. For the ODP value, the diluting ability of chlorofluoromethane ($CCl_3F$, R-11/CFC-11) is used as a reference. Below, in Table 2, are some examples of chemical compounds and their ODP values.

**Table 2.** Chemical compounds and their corresponding ODP values.

| Chemical Compound | Designation | $CCl_3F$ Equivalents |
|---|---|---|
| $CCl_3F$ | R-11 | 1 |
| $CClF_2$-Cl | R-12 | 1 |
| $CClF_2$-F | R-13 | 1 |
| $CClF_2$-H | R-22 | 0.05 |
| $CO_2$ | R-744 | 0 |

### 1.3.3. Acidification Potential

Acidification potential (AP) is a measure of the potential of chemical compounds to acidify soil and water. The calculation of AP is based on $SO_2$ equivalents. Below, in Table 3, are two examples of chemical compounds and their AP values:

**Table 3.** Chemical compounds and their corresponding AP values.

| Chemical Compound | $SO_2$ Equivalents |
|---|---|
| $SO_2$ | 1 |
| $NH_3$ | 1.88 |
| $NO_2$ | 0.7 |

### 1.3.4. Eutrophication Potential

Eutrophication potential (EP) means that an addition of certain fertilizing nutrients in soil and water contributes to algal blooms and bottom death. Nitrogen oxides and phosphates are considered to have a major impact on eutrophication, and the EP value is based on $PO_4^{3-}$ equivalents. Below, in Table 4, are some examples of chemical compounds and their eutrophication potential.

**Table 4.** Chemical compounds and their corresponding EP values.

| Chemical Compound | $PO_4^{3-}$ Equivalents |
|---|---|
| $PO_4^{3-}$ | 1 |
| $NO_2$ | 0.13 |
| $NH_3$ | 0.35 |
| COD | 0.022 |

### 1.3.5. Photochemical Oxidant Creation Potential

Photochemical oxidant creation potential (POCP) is an index to estimate the potential of chemical compounds to form ground-level ozone. Ground-level ozone is directly harmful to humans and nature and is formed when hydrocarbons react together with nitrogen oxides. Nitrogen and hydrocarbon emissions come largely from road traffic. The POCP value is based on $C_2H_4$ equivalents. Listed, in Table 5, below are the POCPs of some chemical compounds.

**Table 5.** Chemical compounds and their corresponding POCP values.

| Chemical Compound | $C_2H_4$ Equivalents |
|---|---|
| $C_2H_4$ | 1 |
| CO | 0.027 |
| $C_2H_6$ | 0.123 |
| $C_7H_8$ | 0.637 |

### 1.3.6. Total Primary Energy

This is the total primary energy use required to manufacture a product. Primary energy is divided into raw materials, i.e., used as material in the product, and energy carriers. Renewable primary energy is biomass, wind power, solar power, and hydropower. Non-renewable primary energy is fossil fuels and uranium.

### *1.4. Purpose of the Paper*

This paper presents a concept study of concrete flooring in a cold climate from a cradle to practical completion (EPD Stages A1–A5) perspective. The study focused on the environmental and material aspects of the dehydration of a concrete floor in a cast-in-place house. The required times for drying and the corresponding environmental impact were

studied for five concrete mixes, with different water-cement ratios, and a case study was conducted for two types of floor slabs in winter conditions, with a maximum drying time limit of 63 days for each floor.

## 2. Materials and Methods

### 2.1. General Description of the Study

In this project, the material and environmental impact of the floors of a cast-in-place concrete house were studied, with a focus on the concrete dehydration required for proper flooring in a cold climate. The study was performed in three stages, as illustrated in Figure 1.

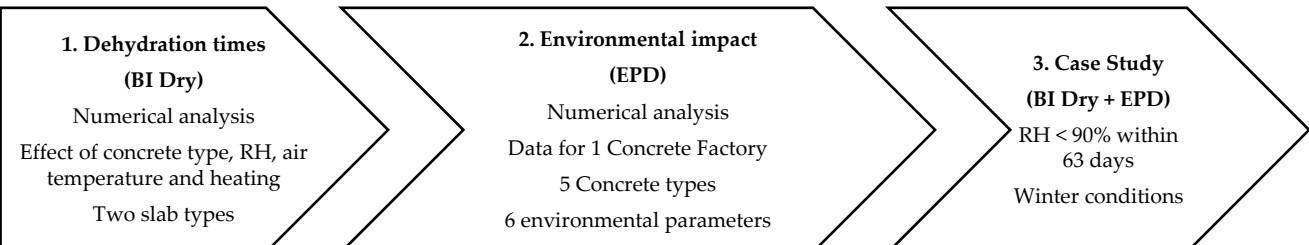

**Figure 1.** The three stages of this study: 1. Parametric study of the dehydration process of concrete floors. 2. EPD-analysis. 3. Case study of the floors of a cast-in-place concrete house, aiming at early carpet installation.

The first stage consisted of a parametric numerical analysis of various factors influencing the dehydration process of concrete floors. All calculations were conducted using the free dehydration calculation software BI Dry [30], and no actual concrete tests were conducted. The concrete floors were divided into two sub-components: ground slab and intermediate floor slabs, described in Sections 2.1.1 and 2.1.2. The second stage focused on the environmental impacts of the concrete options from Step 1 and included environmental product declaration (EPD) calculations for different designs of the concrete floors. The EPD calculations were conducted using the Swedish Concrete Industry Associations Excel-based EPD calculation tool [25]. The third and final stage of the study included a conceptual case study, combining both dehydration (results of Stage 1) and environmental impacts (from Stage 2) of the floors of a cast-in-place concrete house in a cold climate (+5 °C); see Figure 2.

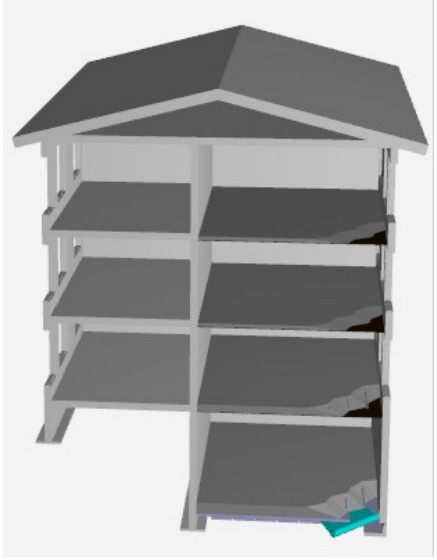

**Figure 2.** Illustration of the conceptual cast-in-place concrete house that was analyzed in the case study. The floor slabs consisted of a ground slab and intermediate slabs.

In the case study, the dehydration procedure was accelerated for faster flooring, and the dehydration criterion was a relative humidity below 90%. The construction project had a maximum limit of 63 days for the dehydration of each floor to avoid delays in the total construction process of the house. The time limit for the dehydration was inspired by a local construction project in the Arctic region. The two alternatives for accelerated dehydration investigated in the case study were:

1.  The use of a fast-dehydrating concrete with a low water-cement ratio.
2.  Heating of the floor slab using cast-in heating cables.

### 2.1.1. Ground Slab Details

The ground floor consisted of a 300 mm concrete slab on top of 150 mm expanded polystyrene (EPS) insulation; see Figure 3. The supporting formwork consisted of traditional plywood sheets, and the ground slab had a total surface area of $A_{GS} = 20 \times 20 = 400$ m$^2$. The total concrete volume of the ground slab was, thus, $V_{GS} = 400 \times 0.3 = 120$ m$^3$, and the corresponding insulation volume was $V_{INS} = 400 \times 0.15 = 60$ m$^3$. In cases where the concrete was heated after casting, heating cables were used with center spacings of s = 100–500 mm, corresponding to a total length of 800–4000 m of cable per floor. The heating cables are typically attached to the reinforcement before casting the concrete

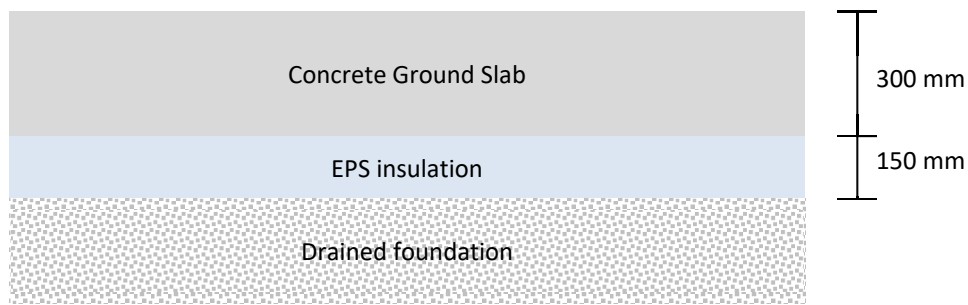

**Figure 3.** Ground slab on top of EPS insulation and drained foundation.

### 2.1.2. Intermediate Slab Details

The intermediate floors consisted of a 200 mm concrete slab that was cast on a traditional concrete form of plywood; see Figure 4. Demolding was carried out 1 week after casting. Each intermediate floor plan had the same surface area as the base plate, i.e., $A_{IS} = 400$ m$^2$, and the concrete volume of each intermediate slab was, thus, $V_{IS} = 400 \times 0.2 = 80$ m$^3$. In cases where heating was required, heating cables with center distance s = 100–500 mm were used, corresponding to a total of 800–4000 m of cable per floor.

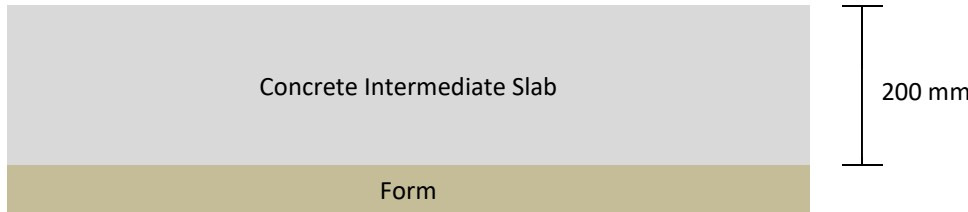

**Figure 4.** Intermediate slab cast on top of a traditional plywood form.

### 2.2. Study of the Dehydration Time

The variations in the drying climate that were studied in the parametric study are summarized in the bullet list below. All concretes were delivered with the same concrete temperature, $T_{concrete}$ = +15 °C. The parameters that were varied and analyzed in the study were the type of slab, the ambient relative humidity (RH), the heating cables' spacing (s), and power (P), as well as the air temperature ($T_{air}$). All parameters were investigated for

five different concrete types (variation in the water-cement ratio). The investigated ranges of each individual parameter were:

- Concrete quality (water-cement ratio 0.34–0.53);
- Air temperature (5–20 °C);
- Humidity (RH 30–80%);
- Type of slab (ground slab and intermediate slabs);
- Heating cable setup (power, $p$ = 0–50 W/m; spacing, s = 0.1–0.5 m).

Drying calculations were carried out by BI Dry, a software developed to calculate drying times for five types of concrete with different water-cement ratios. The cement was of type Bascement, which is a general housing purpose Portland–limestone cement, CEM II/A-LL 42.5 R. Crushed aggregates of two fractions 0–8 and 8–16 mm were included in the mixes, as well as superplasticizers. The concrete in BI-Dry is defined in five strength classes with the properties defined in Table 6.

**Table 6.** Concrete options available in the BI Dry software.

| Concrete | Water-Cement Ratio |
|---|---|
| Concrete 1 | 0.34 |
| Concrete 2 | 0.38 |
| Concrete 3 | 0.43 |
| Concrete 4 | 0.47 |
| Concrete 5 | 0.53 |

The software first calculates the temperature development of the concrete, and the dehydration calculations are based on the temperatures. The heat and moisture models used in the software calculations, as well as the required equations, were reported in [31,32]. The calculations of the heat, maturity, and strength development were based on the results reported in [33].

The requirement for carpet laying was set at a relative humidity in the concrete of <90%, with a safety margin of 2%. The dehydration times were investigated for a winter case and a summer case, with ambient air temperatures of 5 and 20 °C, respectively. For both cases, the dehydration time was investigated for six levels of ambient relative humidity (RH between 30 and 80%) and five levels of heating cable constellations (P between 0 and 500 W/m²). In cases where the floor slab was heated with heating cables for accelerated drying, four power constellations were mainly used:

- 40 W/m²—cables with an output of 20 W/m and a center distance of 0.5 m;
- 100 W/m²—cables with the power 50 W/m and the center distance 0.5 m;
- 200 W/m²—cables with an output of 20 W/m and a center distance of 0.1 m;
- 500 W/m²—cables with an output of 50 W/m and a center distance of 0.1 m.

The analysis plan for the 130 × 2 investigations that were carried out in the parametric dehydration study are summarized in Table 7.

**Table 7.** Plan for the parametric study of dehydration times, including 130 × 2 (intermediate and ground slab) calculations, as seen in the test matrix.

| | Concrete | RH (%) | | | | | | P (W/m²) | | | | |
|---|---|---|---|---|---|---|---|---|---|---|---|---|
| | ($T_{air}$ = 5 °C) | 30 | 40 | 50 | 60 | 70 | 80 | 0 | 40 | 100 | 200 | 500 |
| Winter Case | Concrete 1 | | | | | | | | | | | |
| | Concrete 2 | | | | | | | | | | | |
| | Concrete 3 | | | | | | | | | | | |
| | Concrete 4 | | | | | | | | | | | |
| | Concrete 5 | | | | | | | | | | | |

**Table 7.** *Cont.*

| Concrete ($T_{air}$ = 20 °C) | RH (%) | | | | | | P (W/m$^2$) | | | | |
|---|---|---|---|---|---|---|---|---|---|---|---|
| | 30 | 40 | 50 | 60 | 70 | 80 | 0 | 40 | 100 | 200 | 500 |
| **Summer Case** | | | | | | | | | | | |
| Concrete 1 | | | | | | | | | | | |
| Concrete 2 | | | | | | | | | | | |
| Concrete 3 | | | | | | | | | | | |
| Concrete 4 | | | | | | | | | | | |
| Concrete 5 | | | | | | | | | | | |

| Concrete | $T_{air}$ (°C) | | | |
|---|---|---|---|---|
| | 5 | 10 | 15 | 20 |
| **Temperature Effect** | | | | |
| Concrete 1 | | | | |
| Concrete 2 | | | | |
| Concrete 3 | | | | |
| Concrete 4 | | | | |
| Concrete 5 | | | | |

### 2.3. Study of the Environmental Impact

The environmental impact of the concrete frame was analyzed using The Swedish Concrete Associations Excel-based tool for EPD calculation. The tool is based on a database with environmental impact factors for various input materials in concrete constructions. The analysis in the parameter study was performed for the manufacture of the respective concrete slab and included Modules A1–A5 in its life cycle, i.e., the material production (A1–A3) and construction process (A4–A5) stages. For the case study, the entire concrete frame formed the basis for the EPD analysis. The sub-processes and materials that were included in the EPD analysis are illustrated in Figure 5 and summarized in Table 8.

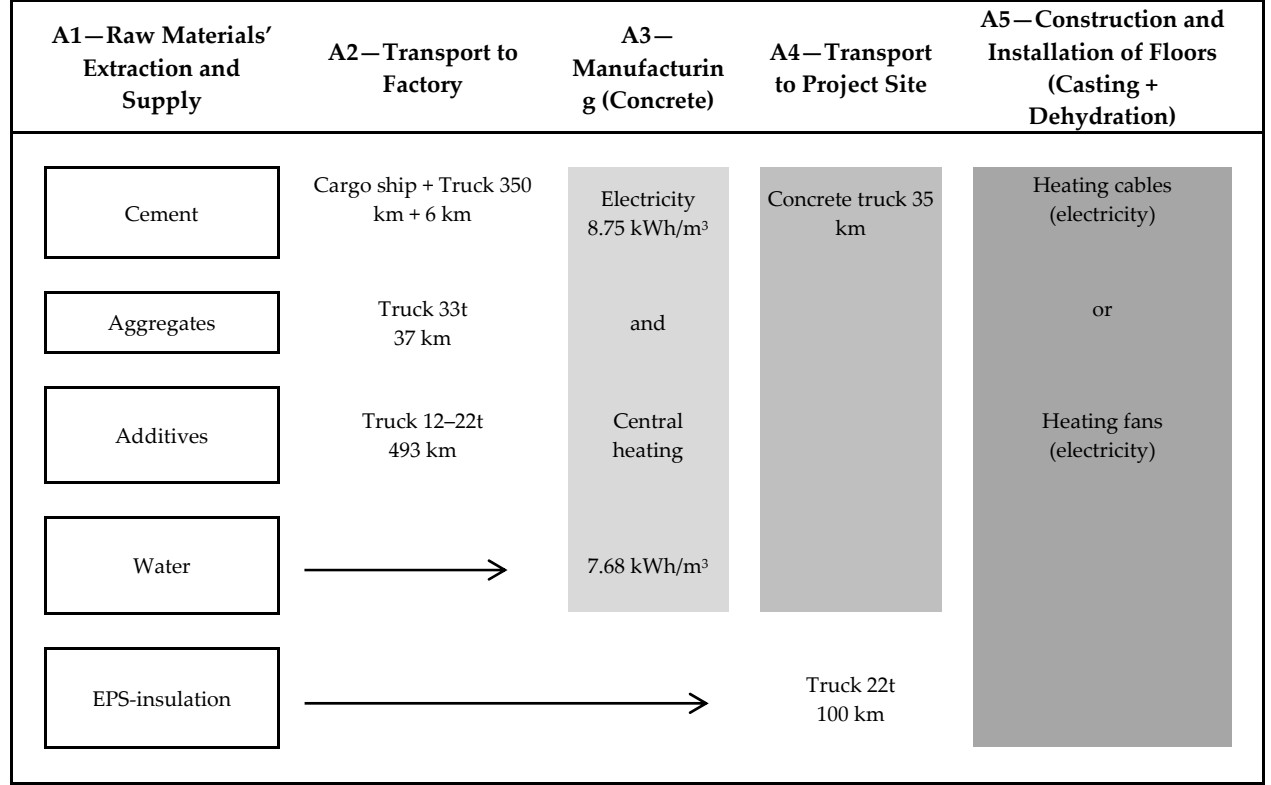

**Figure 5.** Input parameters for the environmental impact calculations. The arrows indicate which Modules are involved for each material and the colors distinguish several combined materials that are affected by the module.

**Table 8.** Description of life cycle Modules A1–A5 included in the environmental study.

| Life Cycle Module | Description |
|---|---|
| A1—Raw material supply | Raw material/input material included in the finished concrete frame. In addition to the concrete components (cement, aggregate, water, and additives), the cellular plastic insulation for the base plate was also included. The EPD parameters were taken from [34–38]. |
| A2—Transportation of raw materials to production facilities | Transport of input material to concrete factory. In this case, all transport could be performed by either cargo ship or truck. The EPD parameters were taken from [39–41]. |
| A3—Manufacturing of construction materials | Manufacturing of concrete. When manufacturing 1 m$^3$ of concrete, approximately 15 kWh of energy is used [42]. In this study, real factory data were used, and the energy for concrete production was roughly half district heating and half electricity. The EPD parameters were taken from [43,44]. |
| A4—Transportation of products to the construction site | Transport of concrete and EPD insulation to the construction site. The concrete was transported by concrete truck (capacity of 6 m$^3$), and the insulation was transported by truck. The transports also included the return distance. |
| A5—Construction and installation | Production of joists. In some cases, electric heating cables or building fans were used to raise the temperature and, thus, speed up the drying process. The EPD parameters were taken from [44]. |

*2.4. Input for Environmental Calculations*

The transport distances and resource consumption to produce concrete were based on site-specific information for an unspecified concrete factory in Sweden, previously reported in [45]. These numbers may vary for different concrete factories, but this study focused only on one factory and gives a comprehensive understanding of the concrete. The means of transportation and transportation distances for the different materials are reported in Figure 5 and Table 9. Energy consumption during production refers to the production of 1 m$^3$ of concrete.

**Table 9.** Summary of transportation distances used for the environmental impact calculations.

| Material | Transportation Vehicle | Transportation Distance |
|---|---|---|
| Cement | Cargo ship, 10,000 dwt, regional | 350 |
| | Truck bulk 22t cargo, SE | 6 km |
| Aggregates | Truck bulk 33t cargo, SE | 37 km |
| Additives | Truck goods 12–22t last, EU | 493 km |
| EPS insulation | Truck bulk 22t last, SE | 100 km |
| Mixed concrete | Concrete truck, 6 m$^3$ | 35 km |

## 3. Results

*3.1. Study of Dehydration Times*

The parameters that were studied regarding their effect on the dehydration times were:

- Concrete quality (Concrete 1–5);
- Type of slab (ground slab or intermediate slab);
- Air temperature (+5, +10, +15, or +20 °C);
- Relative humidity of the air (30, 40, 50, 60, 70, or 80%);
- Impact of heating cables (0, 20 or 50 W/m, with a center spacing of 100 or 500 mm).

The dehydration times for concrete were investigated with the dehydration software, BI Dry, and the results are compiled in Figures 6–9. The figures show the time required for the concrete to dehydrate below the threshold humidity of 90% for the various concretes.

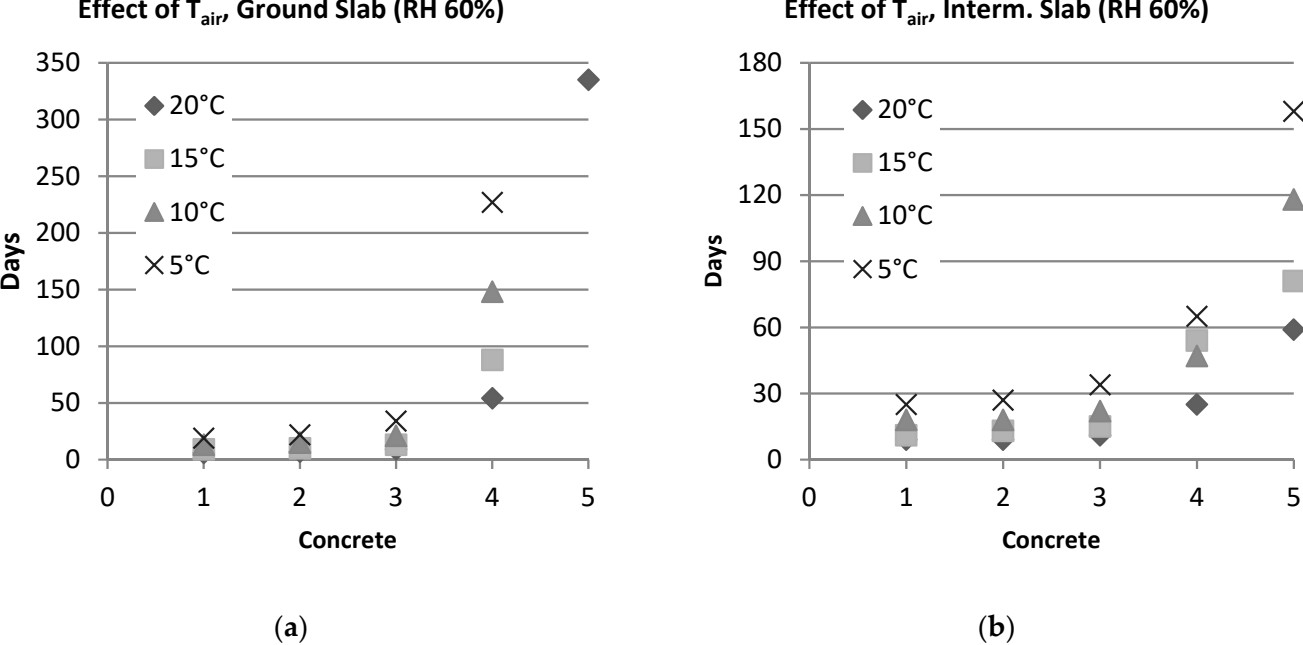

**Figure 6.** Dehydration times to reach a concrete humidity below 90% for a (**a**) ground slab and (**b**) intermediate slab. The results are displayed for five different concrete choices and four ambient air temperatures. The ambient relative humidity was 60% for all cases.

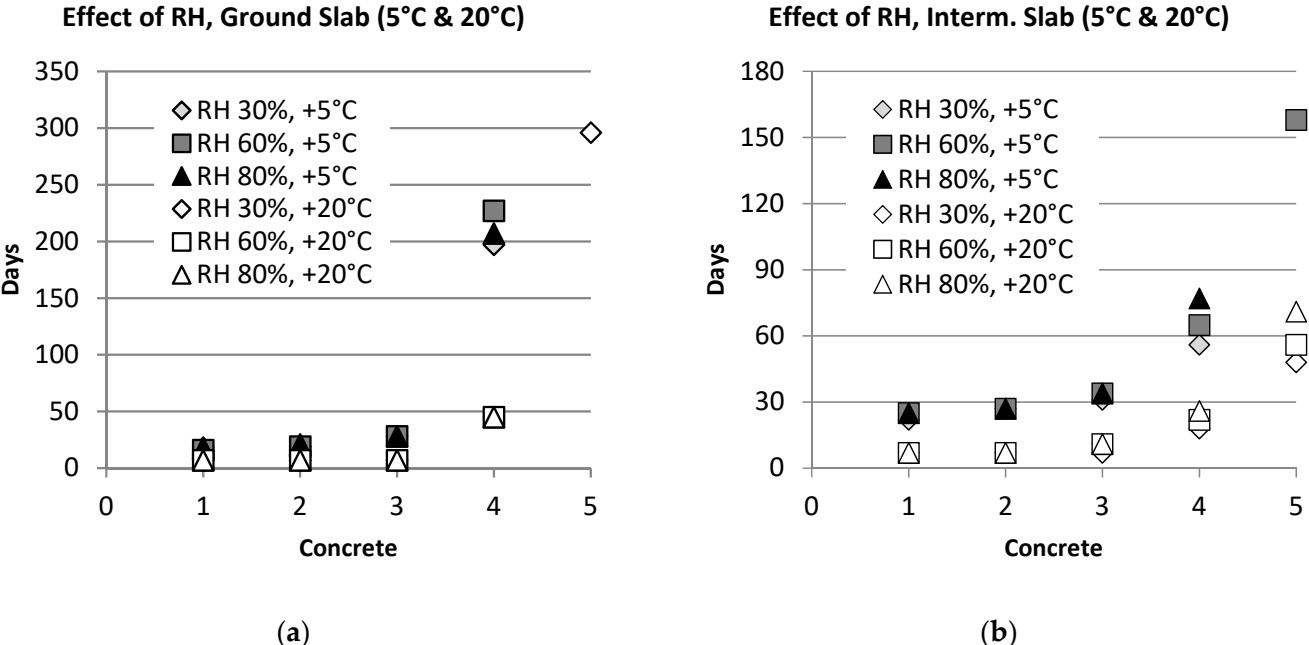

**Figure 7.** Dehydration times to reach a concrete humidity below 90% for a (**a**) ground slab and (**b**) intermediate slab. The results are displayed for five different concrete choices, three levels of ambient relative humidity, and two levels of ambient air temperature.

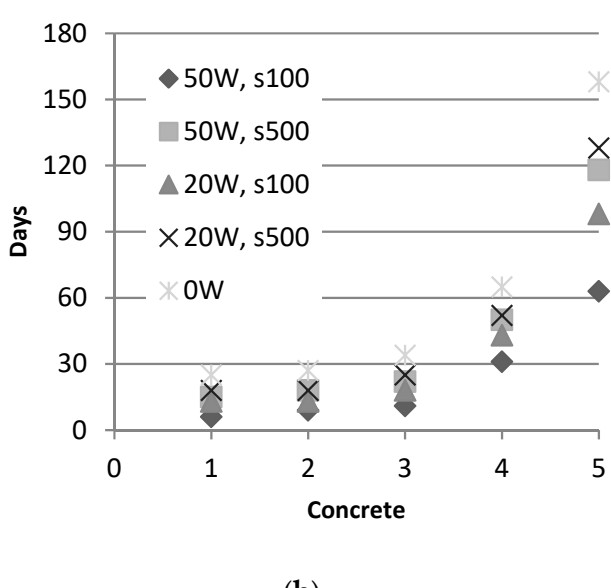

**Figure 8.** Dehydration times to reach a concrete humidity below 90% for (**a**) ground slab and (**b**) intermediate slab. The results are displayed for five different concrete choices and five options of internal heating with heating cables. The ambient relative humidity was 60% for all cases, and the temperature was constant at +5 °C (winter scenario).

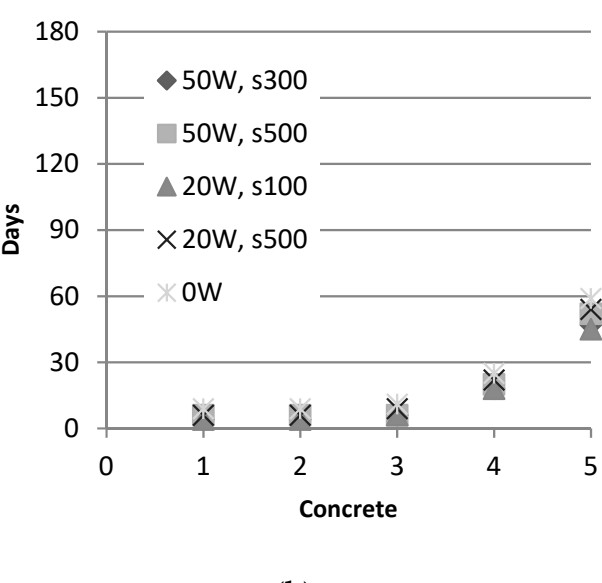

**Figure 9.** Dehydration times to reach a concrete humidity below 90% for a (**a**) ground slab and (**b**) intermediate slab. The results are displayed for five different concrete choices and five options of in-ternal heating with heating cables. The ambient relative humidity was 60% for all cases, and the temperature was constant at +20 °C (summer scenario).

3.1.1. The Air Temperatures Effect on Dehydration Times

Figure 6 shows the dehydration time dependency of the ambient air temperature for a base slab, Figure 6a, and an intermediate slab, Figure 6b. The dehydration time is clearly

dependent on both the air temperature and the concrete quality. A higher air temperature and lower water-cement ratio, respectively, resulted in a faster drying processes.

For example, a ground slab dehydrated within 1 week for Concrete 1 (water-cement ratio = 0.34) if the ambient air temperature was +20 °C and about 2.5 weeks if the air temperature dropped down to +5 °C. The corresponding dehydration time for Concrete 1 and air temperatures of +20 °C and +5 °C were about 1.5 and 3.5 weeks, respectively. Here, we need to consider that the formwork was not removed until after 1 week for the intermediate slabs, which prevented two-sided dehydration in the crucial initial stage of the dehydration process. For Concrete 5, the required dehydration time for a ground slab was 335 days for an ambient temperature of +20 °C, while lower temperatures yielded a dehydration time of almost two years for +15 °C and even longer for lower ambient temperatures. For Concrete 5, the effect of two-sided dehydration for the intermediate slabs was evident. Concrete 5 "only" required about 60 days for an ambient temperature of 20 °C and almost 160 days for an ambient temperature of 5 °C.

### 3.1.2. The Relative Humidity's Effect on Dehydration Times

Figure 7 shows the dehydration time dependency of the ambient relative humidity for a base slab, Figure 7a, and an intermediate slab, Figure 7b. Due to small differences for a 10% change in RH, only three levels (30, 60, and 80%) are shown in Figure 7. For the analysis, a constant ambient temperature of 5 or 20 °C was used (winter case and summer case). The results showed that the dehydration time was less dependent on the ambient relative humidity than the corresponding air temperatures (Figure 6). One result of this parameter study that deviated was that the dehydration of a ground slab and Concrete 4 required a longer time for an ambient relative humidity of 60%, compared to 80%. This could indicate a computational error in the software or that the background equations are not able to correctly incorporate the effect of changes in the ambient relative humidity. The calculations were carried out twice for this case to exclude input errors in the software analysis, but with the same outcome.

### 3.1.3. The Effect of Internal Heating Cables on Dehydration Times

Figures 8 and 9 show the dehydration time dependency of incorporating internal heating cables inside the slabs. For the analysis, constant ambient air temperatures of 5 °C (winter scenario) and 20 °C (summer scenario) were used, respectively, and the air humidity was kept constant at a level of 60%. The results showed, quite logically, that a higher degree of internal heating led to faster dehydration of the concrete.

For the summer scenario, at an air temperature of 20 °C, the BI Dry calculations indicated that heating cables with an output of 50 W/m and a center distance of 100 mm became too hot for the concrete, and in Figure 9, this constellation was, therefore, replaced by heating cables with an output of 50 W/m and a center distance of 300 mm. In any case, the results showed that a higher degree of heating led to faster dehydration. For the intermediate floor, the differences between the different heating options and unheated were obviously very small at an air temperature of 20 °C; see Figure 9.

### 3.2. Study of Environmental Impacts

The results reported in Figures 10–13 are the total environmental impacts of producing 1 m$^3$ of concrete for the five concrete mixes in this study. The results only considered the concrete production stage, i.e., Modules A1–A3, and not the construction stage, Modules A4–A5. The results clearly showed that the environmental impacts were highly related to the cement content of the concrete mix: a higher cement content led to a higher environmental impact. Similar results have been reported previously for concrete, for example by [46–50]. This relation between cement content and environmental impact seemed to be valid for all the six environmental parameters in this study (GWP, energy, ODP, AP, EP, and POCP), shown in Figures 10–12. Concrete 1 had the highest amount of

cement (lowest water-cement ratio); Concrete 5 had the lowest amount of cement; the mixes between are numbered consecutively according to the cement content of the concrete mix.

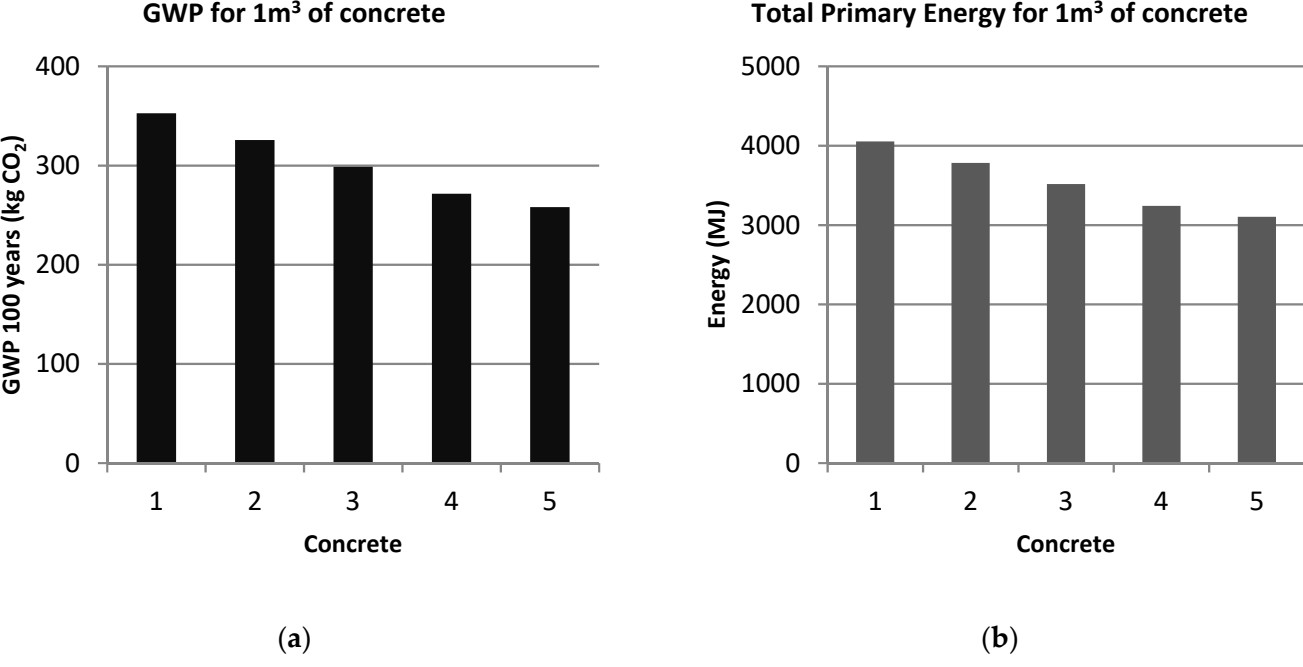

**Figure 10.** (**a**) Global warming potential (GWP) to produce 1 m$^3$ of Concretes 1–5 and (**b**) total primary energy to produce 1 m$^3$ of Concretes 1–5.

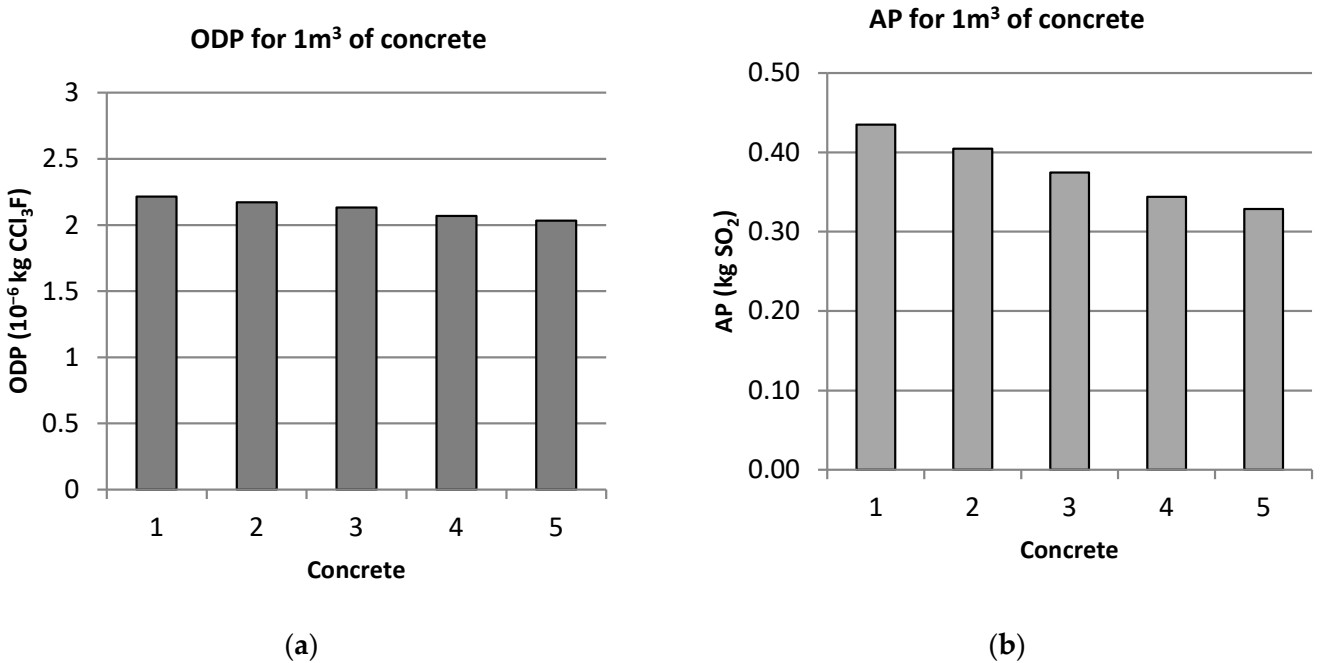

**Figure 11.** (**a**) Ozone depletion potential (ODP) to produce 1 m$^3$ of Concretes 1–5 and (**b**) acidification potential (AP) to produce 1 m$^3$ of Concretes 1–5.

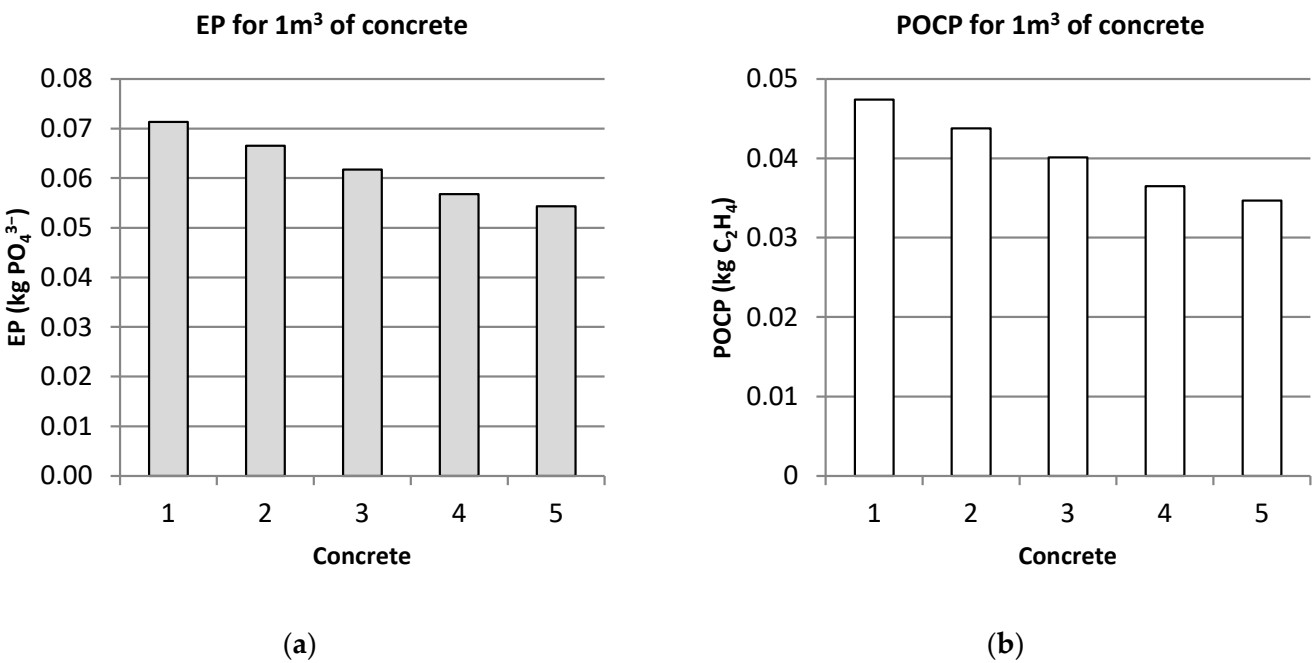

**Figure 12.** (**a**) Eutrophication potential (EP) to produce 1 m$^3$ of Concretes 1–5 and (**b**) photochemical oxidant creation potential (POCP) to produce 1 m$^3$ of Concretes 1–5.

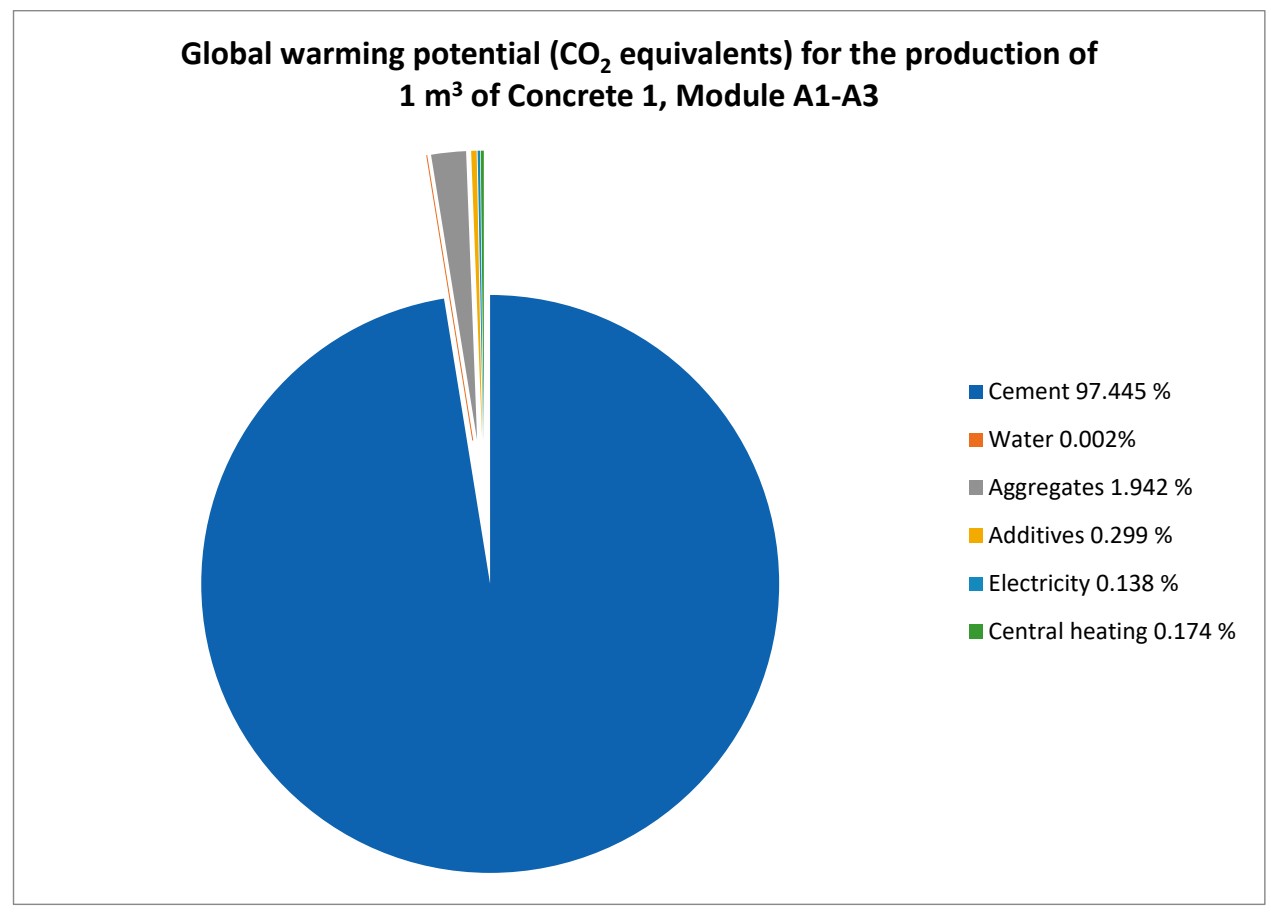

**Figure 13.** Global warming potential ($CO_2$ –equivalents) for 1 m$^3$ of Concrete 1.

Figure 10a shows that the $CO_2$ equivalents, corresponding to the global warming potential, were in the range between 258 kg/m$^3$, for Concrete 5, and 353 kg/m$^3$, for Concrete 1. In other words, Concrete 1 had a 37% higher warming potential, compared to Concrete 5. Concretes 1–5 represent most of the concrete options for the housing industry in Sweden, and the environmental emissions presented in this paper gave a good understanding of the environmental impact of Swedish concrete. Similar results for concrete materials in the northern periphery and Arctic regions were reported in, for example, [12,51,52]. The total primary energy, shown in Figure 10b, ranged between 3103 and 4053 MJ for 1 cubic meter of concrete, a difference of 31%. Figure 11a shows the $CCl_3F$ equivalents (ozone depletion potential) for 1 m$^3$ of concrete, which ranged from 2.0 to 2.2 kg, i.e., a difference of 10%. The $SO_2$ equivalents, corresponding to the acidification potential, were 0.33 kg/m$^3$ for Concrete 1 and 30% higher for Concrete 5, i.e., 0.43 kg/m$^3$, as seen in Figure 11b. The eutrophication potential, shown in Figure 12a, was 0.054 kg $PO_4^{3-}$ per cubic meter of Concrete 5 and 31% higher for Concrete 1. Finally, Figure 12b shows that the photochemical oxidant creation potential ($C_2H_4$ equivalents) was 0.035 kg for 1 m$^3$ of Concrete 5 and 0.047 kg for Concrete 1.

By studying the environmental impact of each material and energy source included in the concrete mix, a strong relation can be identified between environmental impacts and cement content. For the concretes within this study, Concrete 1 had the highest environmental impact for all parameters, while Concrete 5 had the lowest impact. A decreasing linear trend can also be seen for the concretes between Concretes 1 and 5. Figure 13 shows how much each material of Concrete 1 contributes to the GWP of the concrete mix. Cement contributed to almost 97.5% of the total $CO_2$ emissions from the material production of Concrete 1. The second biggest $CO_2$ contributor of Concrete 1 was the aggregates, but they only constituted around 1.9% of the GWP in the material production stage, Modules A1–A3. Similar proportions can be seen also for Concretes 2–5.

## 4. Case Study

Figures 14 and 15 show the environmental impact and the total energy consumption to produce a ground and an intermediate slab of 400 m$^2$, respectively, with the five different concrete options of this study. The slab area corresponds to one floor in a house. The analysis shown in these figures only concerns manufacturing and does not include any measures for accelerated dehydration. The environmental impact is stated as GWP 100 years, i.e., the potential impact of greenhouse gases on the climate over a 100-year period. The impact was calculated according to the IPCC's method, where the greenhouse gases are referred to as the heating capacity of carbon dioxide. The energy consumption is given in GJ. The results showed that a higher water-cement ratio led to a lower climate impact due to the lower amounts of cement and that the GWP was lower for an intermediate slab than a corresponding ground slab due to the smaller floor thickness (200 mm for an intermediate slab and 300 mm for a ground slab).

Figures 16 and 17 show the environmental impact and the total energy consumption to produce a ground and an intermediate slab of 400 m$^2$, respectively, with different concrete options available for this study. The analysis was carried out for a winter case with ambient air temperatures of +5 °C and relative humidity of 60%. For the analysis, there was a maximum limit of 63 days for the dehydration (RH below 90%) of each slab. This requirement resulted in a need for internal heating for both types of slabs with Concrete 4 or 5 being used.

The ground slab with Concrete 4 needed internal heating cables with a total effect of 200 W/m$^2$ for a duration of 63 days to meet the requirements. The ground slab with Concrete 5 needed internal heating cables with a total effect of 500 W/m$^2$ for a duration of 49 days to meet the requirements. The other concrete options (Concrete 1–3) required no internal heating measures to reach the allowed humidity levels within 63 days.

The intermediate slab with Concrete 4 needed internal heating cables with a total effect of 40 W/m$^2$ for a duration of 52 days to meet the requirements. The intermediate slab with Concrete 5 needed internal heating cables with a total effect of 500 W/m$^2$ for a duration of 463 days to meet the requirements. The other concrete options (Concretes 1–3) required no internal heating measures to reach the allowed humidity levels within 63 days.

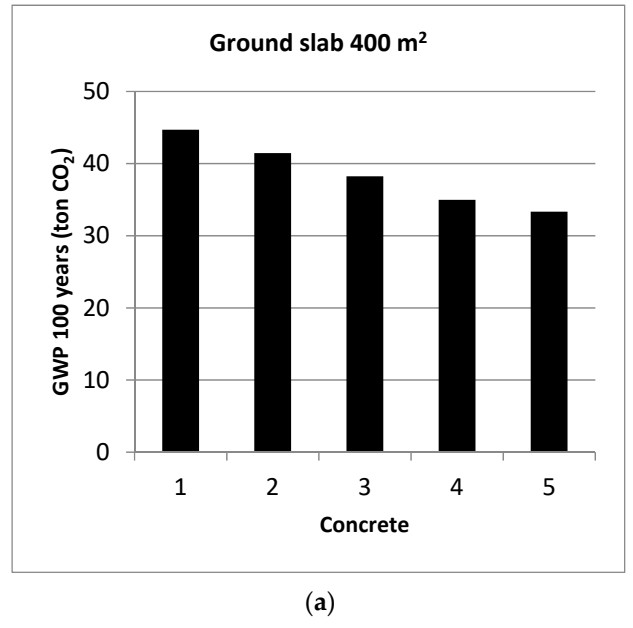

(**a**)

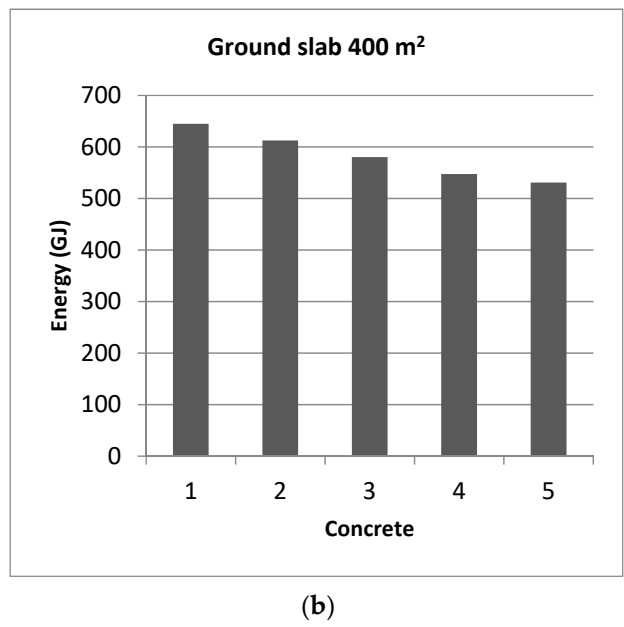

(**b**)

**Figure 14.** (**a**) Global warming potential (GWP) to produce a ground slab with a floor area of 400 m$^2$ of Concretes 1–5 and (**b**) total primary energy to produce a ground slab with a floor area of 400 m$^2$ of Concretes 1–5.

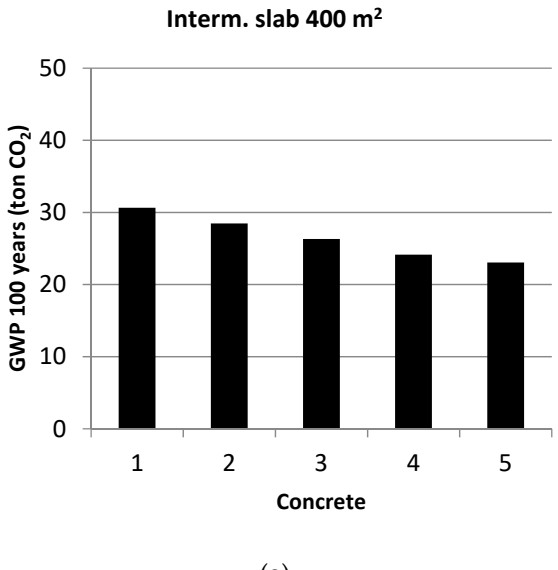

(**a**)

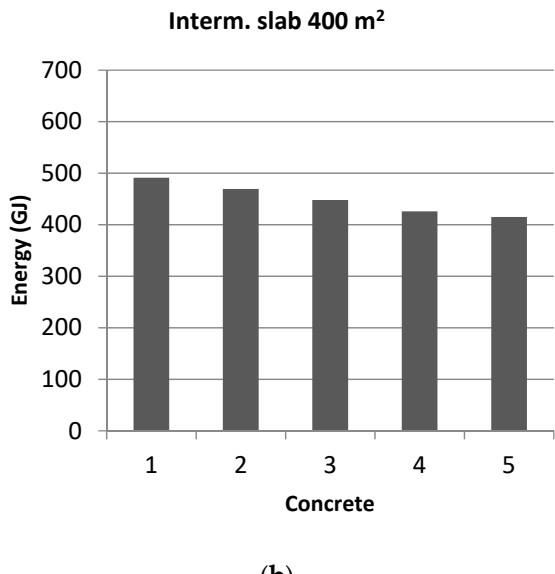

(**b**)

**Figure 15.** (**a**) Global warming potential (GWP) to produce an intermediate slab with a floor area of 400 m$^2$ of Concretes 1–5 and (**b**) total primary energy to produce an intermediate slab with a floor area of 400 m$^2$ of Concretes 1–5.

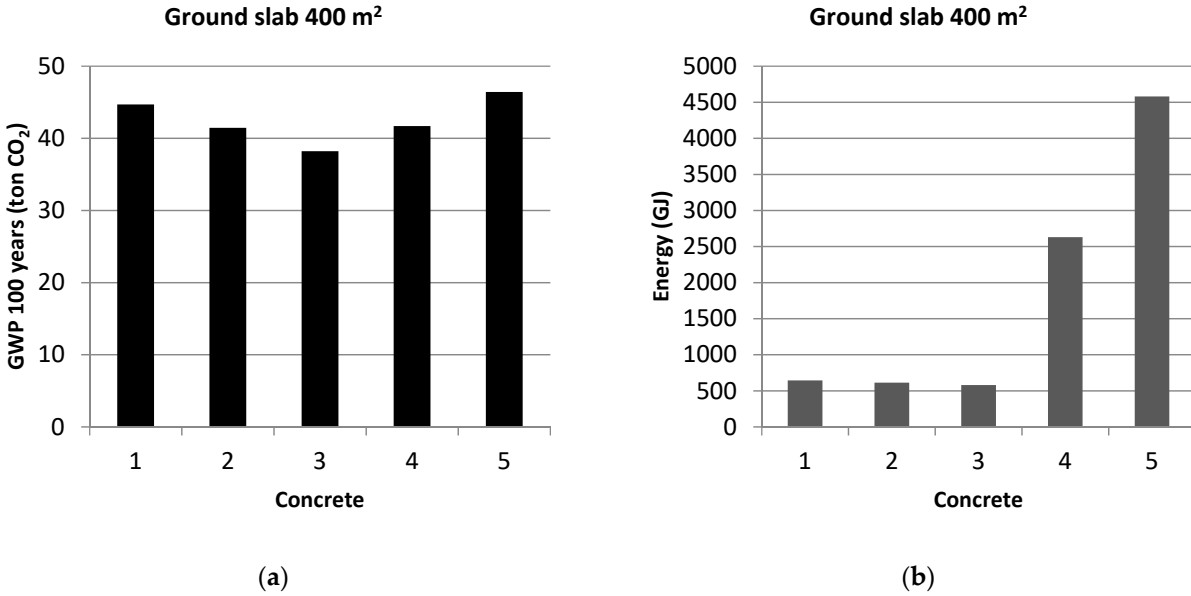

**Figure 16.** (**a**) Global warming potential (GWP) to produce and dehydrate a ground slab with a floor area of 400 m² of Concretes 1–5 within 63 days and (**b**) total primary energy to produce and dehydrate a ground slab with a floor area of 400 m² of Concrete 1–5 within 63 days.

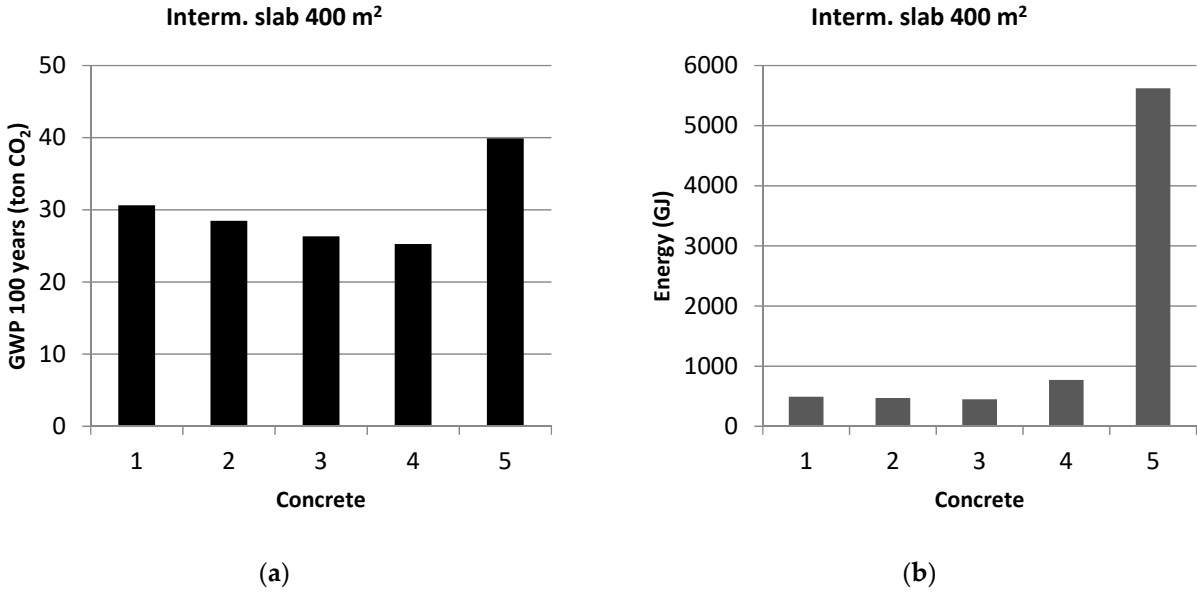

**Figure 17.** (**a**) Global warming potential (GWP) to produce and dehydrate an intermediate slab with a floor area of 400 m² of Concretes 1–5 within 63 days and (**b**) total primary energy to produce and dehydrate an intermediate slab with a floor area of 400 m² of Concretes 1–5 within 63 days.

The results showed that the use of internal heating cables led to an increased environmental impact. The energy consumption was, in fact, multiplied in cases where internal heating cables were needed. This may be due to the large energy and heat loss to the ambient air. One way of making heating cables a more efficient and sustainable solution could be to prevent excessive heat loss to the ambient environment. This could possibly be achieved by different types of installation, but for those kinds of measures, it would be important to sustain efficient ventilation for the evaporating moisture.

## 5. Discussion

There are several methods to accelerate the dehydration and drying process of building structures. Sweden's Construction Industries—Employers' Organization, Sveriges Byggindustrier, for example, recommends three different methods in their handbook *Building Dehydration* [53]: (1) heating, (2) fans, and (3) ventilation, but only the case of heating was investigated in this paper. In this report, a study of parameters affecting the dehydration of concrete were first carried out. All parameters were investigated for five different concrete mixes (variation in the water-cement ratio). Lowering the water-cement ratio is one of the most-common and -simple solutions to reduce the required time for dehydration. This option does not require any additional equipment or installation of heating systems and can be achieved simply by ordering a higher concrete quality to be delivered to the construction site in case of colder weather conditions. A lower water-cement ratio typically results in faster dehydration, but it also affects the environmental impact, as well as the concrete strength and, thereby, the corresponding durability of the structure [54]. Lower water-cement ratios typically indicate a higher amount of cement and, thereby, a higher environmental impact [55]. However, the increased amount of cement results in higher concrete strength and the improved durability of the structure. The strength and durability aspects were, however, not included in the analysis of this paper.

The ranges of each individual parameter, investigated in this paper, were:

- Concrete quality (water-cement ratio 0.34–0.53);
- Ambient air temperature (5–20 °C);
- Ambient relative humidity (RH 30–80%);
- Type of slab (ground slab and intermediate slabs);
- Heating cable setup (power, $P$ = 0–50 W/m; spacing, $s$ = 0.1–0.5 m).

The study showed that the concrete quality, the air temperature, and the constellation of heating cables had a direct effect on the dehydration time. For the cases where the slabs dehydrated to the limit value of 90% relatively quickly (within 1 month), the total dehydration process required less time for the ground slab than for the intermediate slabs. However, for the cases requiring longer dehydration times, the total dehydration process was longer for the ground slab. The intermediate slabs were slightly thinner (200 mm) than the ground slabs (300 mm), and after demolding (7 days after casting), the intermediate slabs were subjected to double-sided dehydration, which theoretically improved the dehydration after demolding. The initial, faster dehydration of the ground slab may be an effect of a higher level of hydration heat developed in the thicker concrete slab [56].

The second stage of the paper focused on the environmental impact and energy consumption within the production stage, Modules A1–A3. The study showed that a lower water-cement ratio led to a lower environmental impact, as well as lower energy consumption. This is a logical outcome due to the lower cement content in concretes with higher water-cement ratios, which has been previously pointed out by several researchers and studies [16,47,57,58]. It also showed that typical concrete options for the Swedish housing industry emits $CO_2$ in a range between 250 and 350 kg/m$^3$ concrete. The primary energy for these concretes ranged between 3100 and 4100 MJ/m$^3$ concrete. The remaining environmental impacts showed the following ranges for typical Swedish housing concretes: the ODP was between 2.0 and 2.2 kg $CCl_3F$/m$^3$; the AP was between 0.33 and 0.43 kg $SO_2$/m$^3$; the EP was between 0.055 and 0.071 kg $PO_4^{3-}$/m$^3$; the POCP was between 0.035 and 0.047 kg $C_2H_4$/m$^3$.

When heating cables were used to accelerate the dehydration process, as investigated in the final stage of the study, the energy consumption increased significantly because of the cable's inefficient electricity consumption. The climate impact in the form of total carbon dioxide emissions was also higher when the dehydration was accelerated by internal heating cables, compared to using self-dehydrating concrete with a lower water-cement ratio, but the difference was not as large. The results showed that, for the cases within

this paper, the most-sustainable solution from an environmental perspective would be to choose a concrete that does not require extra heating for dehydration.

## 6. Conclusions

This paper highlighted the importance of conducting a thorough environmental impact analysis before choosing concrete for a housing project. If only the production stage is considered (Module A1–A3), concretes with high water-cement ratios are undoubtedly the best option, as seen in Figure 10a. However, this study showed that if one is considering a cradle to practical completion perspective, as recommended by the Swedish Housing Agency, a good concrete material (with high water-cement ratio and low $CO_2$ emissions) may require measures that result in even higher total $CO_2$ emissions and energy consumption, compared to concretes containing more cement. Figures 16 and 17 clearly show that concrete with low amounts of cement (Concretes 4 and 5) may end up with higher levels of environmental impact if the analysis includes all stages until practical completion.

The main conclusions of this paper were the following:

- Heating cables can be used to reduce the dehydration time of a concrete slab in a cold climate, but the method is inefficient from an energy perspective.
- The dehydration process can also be accelerated by choosing a lower water-cement ratio in cold climate construction, but this option results in higher environmental impacts.
- A thorough environmental analysis can help to reduce the environmental impact of concrete construction.
- The study supported The Swedish Housing Agency's recommendation that environmental investigations covering the construction stage in the life cycle perspective (Modules A1–A5) are needed to better understand the environmental impact of different construction alternatives.

**Author Contributions:** Conceptualization, J.N.; methodology, J.N.; software, J.N.; validation, J.N. and V.Z.; formal analysis, J.N. and V.Z.; investigation, J.N.; writing—original draft preparation, J.N.; writing—review and editing, J.N. and V.Z.; visualization, J.N. and V.Z. All authors have read and agreed to the published version of the manuscript.

**Funding:** This research was supported by Smart Built Environment U1-2016-07, SBUF, NCC, Betongindustri, Cementa and LTU.

**Data Availability Statement:** All data are available upon request.

**Conflicts of Interest:** The authors declare no conflict of interest.

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
