# Peer review of "Material and Environmental Aspects of Concrete Flooring in Cold Climate"

_constrmater, doi:10.3390/constrmater3020012_

Round 1

Reviewer 1 Report

The work is very interesting and has a lot of data, but it should be better presented.

When reading its conclusion, it is unclear what would be the best procedure to be adopted when executing a structure like the one used in the study.

The paper presents an environmental assessment that, in my view, does not make much sense the way it was presented, since the effects of manufacturing concrete of different strength classes are compared, without taking into account their mechanical performance, reaching a result that, the greater the w/c factor, the better the concrete. However, it is known that this factor affects the durability of the construction. Then the authors should review this analysis.

Author Response

Thank you for the good and thoughtful review. We have addressed all your comments and believe that your suggestions were very helpful to improve the paper.

We have included a discussion about the strength and durability effect of different water-cement ratios. You are absolutely correct that both the strength and durability is improved for a concrete with lower cement content. This was however not included in the analysis, but we hope that the new discussion regarding these aspects improved the paper.

We believe that Table 7 is needed in the paper since it gives illustrative information about the analysis. Research has shown that some people understand illustrative information better than information in text format.

Regarding your comment about Figure 6-9, you are right that some of the results are not possible to distinguish. We tried different styles for Figure 6 to 9, but did not find a better way to show the results. The only option that would help distinguishing between the different points would be to change the scale of the y-axis and thereby exclude some dots for longer dehydration times. We hope that you can consider accepting the current dot-settings and range of the y-axis.

We also separated the discussion and conclusions to make the conclusions more obvious.

We hope that you accept the changes of the paper and that the new version of the paper is a bit clearer for the reader.

Reviewer 2 Report

The article is well written, and the reviewer suggests authors strengthen their discussion part by citing and comparing other similar works. besides, the discussion and conclusion should be separated. 

Some scientific comments:

1, the slab proves a good property in the cold environment. However, even in Sweden, the temperature can be high in summer, then my question is: can the material still show a good performance in high-temperature environment?

2, regarding section 2.1, the authors divide the project into three separate parts. Please show the connection between those three parts. 

3, regarding section 2.1.1, how the ground slab was prepared? Please show more details.

4, regarding the mix. proportion of the concrete, why the five water to cement ratios were selected. Besides, please provide more details of the proportion, such as the chemical composition of cement, the type and size of aggregates, and the type of supplementary cementitious materials (if appliable). 

5, authors only present their own experimental results but do not compare them with other similar studies. It makes the reviewer hard to judge whether the performance of the slab is good or better than others.

Author Response

The article is well written, and the reviewer suggests authors strengthen their discussion part by citing and comparing other similar works. Besides, the discussion and conclusion should be separated.

References have now been included in the discussion to support the findings and conclusions of the paper. The discussion and conclusion have also been separated.

Some scientific comments:

1, the slab proves a good property in the cold environment. However, even in Sweden, the temperature can be high in summer, then my question is: can the material still show a good performance in high-temperature environment?

The concrete investigated for the slab is also suitable for warmer climates and the dehydration-problems are also less important for such conditions.

2, regarding section 2.1, the authors divide the project into three separate parts. Please show the connection between those three parts.

The link between the three stages of the investigation was clarified in the first section of text after Figure 1.

3, regarding section 2.1.1, how the ground slab was prepared? Please show more details.

Additional details on the formwork and installation of heating cables are now provided in the text. No consideration was given to the surface finishing within this study.

4, regarding the mix. proportion of the concrete, why the five water to cement ratios were selected. Besides, please provide more details of the proportion, such as the chemical composition of cement, the type and size of aggregates, and the type of supplementary cementitious materials (if appliable).

Additional details regarding the cement type, aggregates and superplasticizer have been provided. We are unfortunately restricted to provide more details regarding the mix design of the concrete.

5, authors only present their own experimental results but do not compare them with other similar studies. It makes the reviewer hard to judge whether the performance of the slab is good or better than others.

References have now been included in the discussion to support the findings and conclusions of the paper.

We believed that your comments were constructive and provided good support to improve the quality of the paper, and we hope you can accept the revised version.

Reviewer 3 Report

The investigation addresses a trendy and interesting subject. Although the focus of the investigation is centered in a situation occurring in Sweden, it can be extrapolated for other countries/regions with similar climate. The writing is fairly clear and very easy to follow. It is also considered that the manuscript is well-structured. As a general remark, the authors are asked to pay attention to the positioning of figures and tables in the text. Other issues that shall be addressed follow in the annexed file.

Author Response

Thank you for the good and thoughtful review. We have addressed all your comments and believe that your suggestions were very helpful to improve the paper.

The positioning of Figures and Tables have been improved according to your comments and all text-improvements have been introduced.

Regarding your comment regarding the surprising higher total CO2 from electricity compared to central heating. I re-checked the EPD and this number was correct. The "normal" electricity was mainly from a mix of hydro- and wind power with lower CO2 levels than the electricity from central heating.

Regarding your comment regarding setting the 63 day criteria for the calculations. This would have been the best option, but it was not possible for the calculation software we used in the study.

We hope you are happy with our corrections and the improved version of the paper.

Round 2

Reviewer 1 Report

The main suggestions requested were met, thus making the paper eligible for publication.